# Measurement Error Analysis of Seawater Refractive Index: A Measurement Sensor Based on a Position-Sensitive Detector

**DOI:** 10.3390/s24144564

**Published:** 2024-07-14

**Authors:** Guanlong Zhou, Liyan Li, Yan Zhou, Xinyu Chen

**Affiliations:** 1School of Physics, Changchun University of Science and Technology, Changchun 130022, China; zhouguanlong0104@163.com; 2Optoelectronics System Laboratory, Institute of Semiconductors, Chinese Academy of Sciences, Beijing 100083, China; zhouyan@semi.ac.cn

**Keywords:** refractive index measurement sensor, error analysis, position-sensitive detector, optical measurement

## Abstract

The seawater refractive index is an essential parameter in ocean observation, making its high-precision measurement necessary. This can be effectively achieved using a position-sensitive detector-based measurement system. However, in the actual measurement process, the impact of the jitter signal measurement error on the results cannot be ignored. In this study, we theoretically analysed the causes of long jitter signals during seawater refractive index measurements and quantified the influencing factors. Through this analysis, it can be seen that the angle between the two windows in the seawater refractive index measurement area caused a large error in the results, which could be effectively reduced by controlling the angle to within 2.06°. At the same time, the factors affecting the position-sensitive detector’s measurement accuracy were analysed, with changes to the background light, the photosensitive surface’s size, and the working environment’s temperature leading to its reduction. To address the above factors, we first added a 0.9 nm bandwidth, narrow-band filter in front of the detector’s photosensitive surface during system construction to filter out any light other than that from the signal light source. To ensure the seawater refractive index’s measuring range, a position-sensitive detector with a photosensitive surface size of 4 mm × 4 mm was selected; whereas, to reduce the working environment’s temperature variation, we partitioned the measurement system. To validate the testing error range of the optimised test system, standard seawater samples were measured under the same conditions, showing a reduction in the measurement system’s jitter signal from 0.0022 mm to 0.0011 mm, before and after optimisation, respectively, as well as a reduction in the refractive index’s deviation. The experimental results show that the refractive index of seawater was effectively reduced by adjusting the measurement system’s optical path and structure.

## 1. Introduction

The measurement of the seawater refractive index plays a pivotal role in ocean observations [1], with its accuracy being of great importance in research on the seawater flow field, ocean climate prediction, and seabed resource exploration [2,3]. The most recent definition of the thermodynamic equations of seawater (TEOS-10) was proposed by the UNESCO/IOC SCOR/IAPSO Working Group 127 (WG127), and is based on a Gibbs potential function of absolute salinity, temperature, and pressure, focusing on the assessment of absolute salinity [4]. The traditional measurement method for seawater salinity utilises a conductivity sensor to ascertain the seawater conductivity, after which the salinity is calculated using the above-mentioned formula [5,6]. Nevertheless, as the silicates present in the oceans are non-ionic compounds which do not conduct electricity, there is a bias in the measurement of seawater salinity using conductivity sensors [7]. In such a scenario, the use of an optical method for measuring the refractive index is beneficial.

Among the various optical methods for measuring the refractive index of seawater, position-sensitive detector-based seawater refractive index measurement has emerged as a particularly promising approach for high-sensitivity measurements, due to its ability to minimise the influence of environmental factors and its non-contact measurement capabilities. In 2003, Y. Zhao et al. [8] proposed a seawater salinity measurement sensor, designed for the remote monitoring of seawater salinity under laboratory conditions. The resolution of the salinity measurement is 0.012 g/kg, the system’s instability is 0.9%, and the standard deviation of the corresponding seawater refractive index measurement is approximately 10^−5^ RIU. In 2009, a sensor for the in situ detection of the seawater refractive index was presented by D. Malardé et al. [9] The resolution of the seawater refractive index is $4\times 10^{-7}\;RIU$. The standard deviation of seawater refractive index measurements at standard atmospheric pressure and at 35‰ salinity is $4.7\times 10^{-6}\;RIU$. A novel optical measurement structure, based on a position-sensitive detector, has recently been developed, with the objective of enabling higher-sensitivity seawater refractive index measurements in the order of 10^−9^ RIU. In 2024, G. Zhou et al. [10] proposed a highly sensitive seawater refractive index measurement method based on a position-sensitive detector (PSD). The sensitivity of the seawater refractive index measurement of this system is $9.93\times 10^{-9}\;RIU$, and the standard deviation of the refractive index is $\pm 7.54\times 10^{-9}\;RIU$. Nevertheless, several additional factors may contribute to seawater refractive index measurement inaccuracies when employing a position-sensitive detector, including the optical path configuration within the measurement region, system temperature, detector-generated noise, and signal processing-associated uncertainty. While developing highly sensitive seawater refractive index measurement techniques, researchers have striven to eliminate these measurement errors [11].

The current research focus in the field of position-sensitive detector-based measurement systems is the analysis of and reduction in the influence of various factors—including the detector’s output response linearity [12], the injection current [13], and the laser spot size [14]—on their accuracy. In the event of erroneous measurement results caused by jitter signals, compensation systems are frequently employed to obtain more accurate measurement signals [15]. In this context, more effective solutions must be developed to achieve more accurate measurements while maintaining optimal position-sensitive detector conditions.

Based on these new application requirements, an analysis was conducted to identify the underlying causes of these errors, specifically regarding the existing PSD-based seawater refractive index measurement system. Firstly, a theoretical analysis was conducted to identify the error factors generated during optical path transmission based on a position-sensitive detector. To ensure the accuracy of seawater refractive index gradient measurements, it is essential to guarantee that the laser incidence and receiving windows, which collectively constitute the measurement zone, are perfectly parallel, and that the laser itself is perpendicularly incident upon this zone. Consequently, the impact of the angle between the two windows and the laser incidence angle on measurement sensitivity were analysed in this study. The findings indicate that both angles should be maintained within a narrow range of 2.06° to ensure uncompromised measurement sensitivity. Then, a theoretical analysis was conducted to investigate the factors influencing the measurement system’s internal environment, revealing that the interior was affected by the temperature and atmospheric pressure. Based on the results of this analysis, an experimental study was carried out to verify the findings, alongside an analysis aiming to identify the factors influencing detector performance. Based on the above, an improvement plan was developed to address the errors caused by these factors to ensure optimal position-sensitive detector operation. Finally, an experimental comparison and analysis were conducted to assess the stability of the measurement system before and after reducing the error factor, demonstrating a reduction in the jitter signal from ±0.0022 mm to ±0.0011 mm, a reduction in the refractive index deviation of $1.09\times 10^{-8}\;RIU$, and a reduction in the standard deviation from $10^{-8}$ to $10^{-10}\;RIU$. The experimental results show that the seawater refractive index error was effectively reduced by adjusting the measurement system’s optical path and structure.

## 2. The Principle of the Refractive Index Optical Measurement System

Optical seawater refractive index measurements adopt the laser beam deviation technique, and the schematic diagram of a position-sensitive detector-based system is shown in Figure 1.

The light emitted by the laser passes through the incidence window and vertically reaches the area where the seawater refractive index is measured. The seawater refractive index gradient gives rise to a refraction phenomenon when the laser interacts with the seawater, after which the laser traverses the receiving window, where refraction occurs because of the light refraction law. The light is focused by the lens on a single point on the PSD’s photosensitive surface, at which point the coordinates of this laser spot are determined by the PSD.

The refractive index gradient is present in the *y*-direction of the refractive index measurement area. Consequently, when the laser interacts with the seawater within the measurement area, it is influenced by the refractive index gradient, resulting in laser deflection. Therefore, within the theoretical framework of seawater refractive index measurement, only the refractive index gradient in the *y*-direction is considered. The curvature of the refracted light can be expressed as follows [16]:(1)$$\frac{\partial^{2}y}{\partial z^{2}}=\frac{1}{n}\frac{\partial n}{\partial y}$$
where the curvature of the refracted light is $\frac{\partial^{2}y}{\partial z^{2}}$, the light propagation direction is z, the refractive index is *n*, and the gradient of the refractive index of seawater is $\frac{\partial n}{\partial y}$.

The refractive index of seawater, as determined by the PSD-based seawater refractive index measurement system, can be expressed as follows [10]:(2)$$n=n_{0}+\delta_{y}\frac{D_{y}}{fL}$$
where the refractive index of the surrounding seawater is *n*_0_, the displacement of the light rays in the direction of the *y*-axis is $\delta_{y}$, the displacement on the photosensitive surface of the PSD is $D_{y}$, the focal length of the focusing lens is *f*, and the distance between the laser incidence and receiving windows is *L*.

As demonstrated by Equation (2), the sensitivity of a position-sensitive detector-based seawater refractive index measurement system is contingent upon the length of the measurement zone, the focusing lens’ focal length, and the displacement on the PSD’s photosensitive surface. Therefore, the angle between the two windows that comprise the refractive index measurement area can result in the refraction of light during optical path transmission, which, in turn, can lead to a displacement deviation on the PSD’s photosensitive surface, subsequently giving rise to an error in the results.

## 3. A Study of the Errors Induced by the Seawater Refractive Index Measurement System’s Optical Path

During the transmission of the seawater refractive index measurement system’s optical path, the angle between the laser incidence and receiving windows that comprise the measurement area results in a laser light refraction phenomenon, leading to a measurement error. The optical path transmission process is depicted in Figure 2, with the laser incident at a right angle to the main optical axis. The first, second, third, and fourth refractions occur, respectively, from the air medium through the first glass surface of the laser incidence window, from this window to the seawater refractive index measurement area, from this area through the first surface of the laser receiving window, and, finally, from the glass of the laser receiving window to the air medium. Following these four refractions, the laser reaches the photosensitive surface of the PSD, where the position coordinates of the laser spot are obtained.

The light refraction law indicates that the angle of incidence α of a laser is refracted by the refractive index $n_{a}$ of the air medium and the refractive index $n_{g}$ of the window glass, resulting in the refractive angle $i_{1}$, expressed as follows:(3)$$n_{a}\sin\alpha =n_{g}\sin i_{1}$$

The laser is incident at an angle of incidence $i_{1}$ from the window glass, with a refractive index $n_{g},$ to the seawater refractive index measurement area, with a refractive index $n_{s}$, producing an angle of refraction of $2\alpha -i_{2},$ and the following refractive relation:(4)$$n_{g}\sin i_{1}=n_{s}\sin\left(2\alpha -i_{2}\right)$$

The laser is incident at an angle of incidence $i_{2}$ from the seawater refractive index measurement area, with a refractive index of $n_{s},$ to the window glass, with a refractive index of $n_{g}$, producing an angle of refraction of $i_{3}$, expressed as follows:(5)$$n_{s}\sin i_{2}=n_{g}\sin i_{3}$$

The laser is incident at an angle of incidence $i_{3}$ from the window glass, with a refractive index of $n_{g},$ to an air medium with a refractive index of $n_{a}$, producing an angle of refraction of A, expressed as follows:(6)$$n_{g}\sin i_{3}=n_{a}\sin A$$

Equation (7) can be obtained from Equations (3) and (4):(7)$$n_{a}\sin\alpha =n_{s}\sin\left(2\alpha -i_{2}\right)$$

Equation (8) can be obtained from Equations (5) and (6):(8)$$n_{s}\sin i_{2}=n_{a}\sin A$$

The differentiation of Equations (7) and (8) can be obtained as follows:(9)$${\Delta n}_{s}\Delta i_{2}=\Delta n_{s}\tan\left(2\alpha -i_{2}\right)$$
(10)$$\Delta n_{s}\sin i_{2}+n_{s}\Delta i_{2}\cos i_{2}=n_{a}\Delta A\cos A$$

The substitution of Equation (9) into Equation (10) and their subsequent collation yield the following equations:(11)$$\Delta n_{s}=\frac{n_{a}\Delta A\cos A}{\sin i_{2}+\cos i_{2}\tan\left(2\alpha -i_{2}\right)}$$
(12)$$i_{1}=arc\sin\left(\frac{n_{a}\sin\alpha }{n_{g}}\right)$$
(13)$$i_{2}=2\alpha -arc\sin\left(\frac{n_{a}\sin\alpha }{n_{s}}\right)$$
(14)$$i_{3}=arc\sin\left(\frac{n_{s}\sin i_{2}}{n_{g}}\right)$$
(15)$$\cos A=\sqrt{1-{\left(\frac{n_{g}\sin i_{3}}{n_{a}}\right)}^{2}}$$

When the angle between the two windows is 2α and the length of the seawater refractive index measurement zone is $L_{1}$, the laser is incident at position $d_{1}$ of the laser incidence window opening, which is determined using the following trigonometric relationship:(16)$$d_{1}=\frac{2\sin\alpha }{L_{1}}$$

The position designated $d_{2}$, where the laser exits the laser incidence window, is defined as follows:(17)$$d_{2}=d_{1}-t_{i}\tan i_{1}$$
where $t_{i}$ is the glass thickness of the laser incidence window.

The laser light’s position upon incidence onto the laser receiving window following its passage through the seawater refractive index measurement zone is $d_{3},$ and is expressed as follows:(18)$$d_{3}=d_{2}\left(\cos2\alpha +\sin2\alpha \;tani_{2}\right)$$

Position $d_{4}$, where the laser exits the laser receiving window, is defined as follows:(19)$$d_{4}=d_{3}+t_{r}\tan i_{3}$$
where $t_{r}$ is the glass thickness of the laser receiving window.

Position *D*, the spot on the photosensitive surface where the laser is received, can be expressed as follows:(20)$$D=d_{4}\cos\alpha +\left({L_{2}-d}_{4}\sin\alpha \right)\tan\left(A-\alpha \right)$$
where $L_{2}$ represents the distance between the apex of the second surface of the laser receiving window and the PSD’s photosensitive surface.

The differentiation of Equation (20) yields the following:(21)$$\Delta D=\Delta d_{4}\cos\alpha -\Delta d_{4}\sin\alpha \tan\left(A-\alpha \right)+\left({L_{2}-d}_{4}\sin\alpha \right)\Delta \tan\left(A-\alpha \right)$$

Therefore, $\frac{\Delta D}{\Delta n_{s}}$ can be expressed as follows:(22)$$\frac{\Delta D}{\Delta n_{s}}=\frac{\Delta d_{4}\cos\alpha -\Delta d_{4}\sin\alpha \tan\left(A-\alpha \right)+\left({L_{2}-d}_{4}\sin\alpha \right)\Delta \tan\left(A-\alpha \right)}{n_{a}\Delta A\cos A}\times \frac{\sin i_{2}+\cos i_{2}\tan\left(2\alpha -i_{2}\right)}{n_{a}\Delta A\cos A}$$
(23)$$\Delta \tan\left(A-\alpha \right)=\frac{2\Delta A}{1+\cos\left(2A-2\alpha \right)}$$

The relationship between the seawater refractive index and the angle between the two glass windows in the measurement area is shown in Figure 3. According to what is known about the refractive index optical measurement system and the selection of k9 glass for the windows comprising the measurement area, the glass windows’ refractive index is $n_{g}=1.5168$ and their thickness is $t_{i}=t_{r}=2\;mm$. The length of the measurement area is $L_{1}=100\;mm$, and the distance between the apex of the second surface of the laser receiving window and the PSD’s photosensitive surface is $L_{2}=300\;mm$. The air’s refractive index under ambient conditions, standard atmospheric pressure, and a temperature of 25 °C is $n_{a}=1.0003;$ and the initial seawater refractive index, at a salinity of 35 psu, is $n_{s}=1.3381096$. The PSD’s sensitivity is $\Delta D=10^{-3}\;mm$, and the angle between the two glass windows in the measurement area is set to 0.1°≤α≤4°.

Given that the PSD-based optical measurement system for the seawater refractive index has a sensitivity of $9.93\times 10^{-9}\;RIU$, it is crucial to regulate the measurement error resulting from an angle smaller than $9.93\times 10^{-9}\;RIU$ between the two glass windows, to ensure that this error range is satisfied when the angle is 2α<2.06°. It can be demonstrated that the measurement error range is satisfied when $\frac{\Delta D}{\Delta n_{s}}<\frac{107.875\;\mu m}{\mu RIU}$, and the refractive index’s measurement sensitivity changes by $3\times 10^{-10}\;RIU$ for every 0.01° of change in the measurement area’s angle.

## 4. Noise and Measurement Error Analyses of the Position-Sensitive Detector

The operational principle of the PSD’s measurement position is shown in Figure 4. When the laser irradiates the PSD’s photosensitive surface, electrons are excited from the valence band to the conduction band through the photovoltaic effect, thus generating a photocurrent. The PSD has a PIN structure: its light-sensitive surface is the uniformly distributed resistive P layer; the N layer is connected to the common electrode; and the electrodes on both sides are used to extract positional signals.

When the PSD’s centre is taken as the reference point, the laser spot irradiates the photosensitive surface, resulting in electrodes 1 and 2 generating corresponding photocurrents. The relationship between the electrodes’ output currents and the incidence spot’s position can be expressed as follows.

The output currents of electrodes 1 and 2 are, respectively, the following:(24)$$I_{1}=I_{0}\frac{\frac{L_{Y}}{2}-Y_{A}}{L_{Y}}$$
(25)$$I_{2}=I_{0}\frac{Y_{B}}{L_{Y}}$$
where $I_{1}$ is electrode 1’s output current, $I_{2}$ is electrode 2’s output current, $I_{0}$ is the total output photocurrent of electrodes 1 and 2, $L_{Y}$ is the photosensitive surface’s length, $Y_{A}$ is the distance from the incident laser spot position to the centre of the photosensitive surface, and $Y_{B}$ is the distance from the incident laser spot position to electrode 1.

The positional resolution of the PSD is determined by the resistor length $L_{Y}$ and the signal-to-noise ratio, establishing the following equation according to Equation (25):(26)$$\Delta Y=\frac{\Delta IL_{Y}}{I_{0}}$$
where ΔI is the output current change amount, and ΔY is the tiny displacement.

In the case of infinitely small position displacements, the amount of noise contained within output current signal $I_{2}$ determines the position resolution. If the noise current of the PSD is $I_{n}$, its position resolution ΔR can be expressed as follows:(27)$$\Delta R=\frac{I_{n}L_{Y}}{I_{0}}$$

According to Equation (27), the PSD’s position resolution can be improved by decreasing the photosensitive surface’s length $L_{Y},$ or by increasing the signal photocurrent. The PSD also has non-linear effects, such as an output current from electrodes at both ends, the incidence spot’s light intensity, characteristic differences in signal amplification and processing in the subsequent processing circuit, and A/D conversion errors. Segmented optimisation BP networks and multilayer conjugate gradient optimisation algorithms are applied for a non-linear correction.

Noise exists in the PSD during operation, so its shot and thermal noise and operational amplifiers are analysed according to the PSD’s noise equivalent model, while the root mean square of the total noise voltage is calculated considering the feedback resistor.

The PSD’s shot noise mainly originates from the photocurrent and dark current and can be expressed as follows:(28)$$I_{s}=\sqrt{2q\times \left(I_{0}+I_{D}\right)\times B}$$
where q is the electronic charge, $I_{D}$ is the dark current, and B is the bandwidth.

When the ratio between the interpole and feedback resistance is greater than 0.1, the latter is non-negligible, and the shot noise output voltage $V_{s}$ is as follows:(29)$$V_{s}=R_{f}\sqrt{2q\times \left(I_{0}+I_{D}\right)\times B}$$
where $R_{f}$ is the feedback resistance.

In the PSD, the random movement of charge carriers in the resistive material, due to temperature changes, causes thermal noise. The thermal noise current $I_{j}$ generated by the interpole resistance is as follows:(30)$$I_{j}=\sqrt{\frac{4kTB}{R_{ie}}}$$
where k is Boltzmann’s constant, T is the absolute temperature at which the PSD operates, and $R_{ie}$ is the inter-electrode resistance.

When the ratio between the interpole and feedback resistance is greater than 0.1, the latter is non-negligible, and the thermal noise output voltage $V_{j}$ generated by the resistance between the poles can be expressed as follows:(31)$$V_{j}=R_{f}\sqrt{\frac{4kTB}{R_{ie}}}$$

The operational amplifier equivalent noise input current $I_{en}$ can be expressed as follows:(32)$$I_{en}=\frac{V_{e}\sqrt{B}}{R_{ie}}$$
where $V_{e}$ is the noise input voltage of the operational amplifier.

When the ratio between the interpole and feedback resistance is greater than 0.1, the latter is non-negligible, and the noise generated by the operational amplifier’s equivalent noise input voltage corresponds to the output voltage $V_{en}$, denoted as follows:(33)$$V_{en}=\left(1+\frac{R_{f}}{R_{ie}}\right)I_{en}\sqrt{B}$$

The voltage $V_{Rf}$, corresponding to the thermal noise of the feedback resistor, is expressed as follows:(34)$$V_{Rf}=R_{f}\sqrt{\frac{4kTB}{R_{ie}}}$$

The operational amplifier’s equivalent noise voltage $V_{in}$ is expressed as follows:(35)$$V_{in}=R_{f}I_{n}\sqrt{B}$$
where the PSD’s noise current RMS value $I_{n}$ is expressed as
(36)$$I_{n}=\sqrt{I_{s}^{2}+I_{j}^{2}+I_{e}^{2}}$$

Therefore, the RMS value of the PSD noise voltage due to the operational amplifier equivalent noise input is expressed as follows:(37)$$V_{n}=\sqrt{V_{s}^{2}+V_{j}^{2}+V_{en}^{2}+V_{Rf}^{2}+V_{in}^{2}}$$

According to the PSD used in the existing seawater refractive index measurement system, when the bias voltage and load resistance are 5 V and 1 kΩ, respectively, the corresponding saturation photocurrent $I_{n}$ and bandwidth B are 10^−4^ A and 10 Hz. In the operational amplifier, the feedback resistor $R_{f}$ is 100 kΩ, the typical value of the interpolar resistor $R_{ie}$ is 50 kΩ, the equivalent noise input voltage $V_{e}$ is 0.003 V, the electron charge q is $1.6\times 10^{-19}\;C$, and Boltzmann’s constant k is $1.38\times 10^{-23}\;J/K$. Figure 5a represents the curve of the PSD’s noise current with a temperature change from 20 to 25 °C, which increases gradually with the folding temperature. Figure 5b represents the PSD’s noise voltage with an operating temperature change curve from 20 to 25 °C, which gradually increases with an increase in the fold temperature. Figure 5c represents the PSD’s position resolution with an operating temperature change curve from 20 to 25 °C. Since the PSD’s photosensitive side is 4 mm long and, according to Equation (27), an increase in the folding temperature results in the PSD’s position resolution gradually increasing, this leads to a reduction in the PSD’s measurement accuracy.

At the same time, there are some factors that can affect the PSD’s measurement accuracy of the seawater refractive index. For instance, the presence of background light at the laser incidence spot on the PSD’s photosensitive surface results in a position measurement error. Therefore, in our study, a filter is added to the PSD’s photosensitive surface to filter out any background light other than that of the signal light. In addition, the size of the incidence spot on the PSD’s photosensitive surface directly affects the PSD’s positioning repeatability, which is bound to be reduced when the spot is large, since the photocurrent is proportional to the irradiance on the PSD. In our measurement system, a focusing lens is used to converge the light spots, thus ensuring the PSD’s positioning repeatability. In the measurement system, it is difficult for the beam’s symmetry axis to always be parallel to the normal axis of the PSD’s photosensitive surface, so the projection of the circular spot on this surface is elliptical and equivalent to a large spot, necessarily reducing the repeatability of the PSD’s positioning.

## 5. Structural Optimisation of a PSD-Based Seawater Refractive Index Measurement System

The seawater refractive index optical measurement system is based on a PSD (TEM Messtechnik GmbH, Hannover, German), and its optimised optical path structure is shown in Figure 6.

Due to the laser’s propagation in seawater, there were light absorption and scattering phenomena; so, after a comprehensive consideration of seawater’s attenuation of light, we chose a 532 nm laser (CPS532-C2; Thorlabs, Newton, NJ, USA) wavelength as the signal light source. To ensure that the two glass windows in the measurement area remained parallel, and that the laser was parallel incident to the main optical axis, the optical path of the measurement system was adjusted using a parallel light pipe. To reduce optical path propagation errors, the number of optical lenses was reduced while maintaining the refractive index measurement sensitivity. To avoid position measurement errors caused by ambient background light, a narrow-band filter with the same peak wavelength and a 0.9 nm bandwidth was placed in front of the PSD photosensor to remove any stray light outside of the 532 nm measurement light. To improve the PSD’s measurement accuracy and ensure the measurement range of the seawater refractive index, a position-sensitive detector with a 4 mm×4 mm photosensitive surface was selected as the measurement sensor. During the refractive index measurement process, the PSD’s thermal noise and dark current changes with the operating temperature, resulting in differences in measurement accuracy. Therefore, during the structural design of the seawater refractive index measurement system, the signal processing module in the PSD that causes the temperature to change was divided into two parts, an optical and an electrical section, to reduce the variation in the optical path and the PSD’s operating temperature. A physical version of the optimised seawater refractive index measurement system is shown in Figure 7.

## 6. Stability Experiment on the Seawater Refractive Index Measurement System

### 6.1. Experimental Setup

The stability of a seawater refractive index measurement system is an important indicator of reliability, reflecting the accuracy of its measurements. The stability experiment was conducted in an indoor, glass water cylinder with constant temperature and pressure levels, as illustrated in Figure 8. The measurement zone of the refractive index measurement system was immersed in a glass tank containing a standard seawater solution provided by the National Centre for Marine Standards and Metrology of China. The refractive index of the standard seawater solution, with a salinity of 35.003 PSU, was 1.3381096 when the temperature of the measurement environment was 26 ℃ and the standard atmospheric pressure was 10.1325 dbar. The relevant parameters of the standard seawater solution used in the stability experiments are listed in Table 1.

### 6.2. Experimental Stability Results

In the experiments, the system employed a sampling time of 60 s, with a sampling frequency of 10 kHz. Figure 9a depicts the position signal measured by the system prior to the implementation of structural partitioning for the purpose of regulating the PSD’s operational temperature. The figure illustrates an average value of −0.0571 mm. A calculation was performed to determine the refractive index measurement deviation before the optimisation of the temperature control structure of the measurement system. This was performed by taking into account the jitter signal, which was observed to be ±0.0022 mm, and the seawater refractive index measurement sensitivity of $9.93\times 10^{-9}\;RIU$. The result of this calculation was $\pm 1.19\times 10^{-8}\;RIU$. Following the introduction of structural partitioning for the purpose of regulating the PSD’s operational temperature, the position signal measured by the system is depicted in Figure 9b, exhibiting an average value of −0.0575 mm, and the system’s jitter signal was ±0.0018 mm. In conjunction with the sensitivity of the measurement system, a refractive index deviation of $\pm 7.944\times 10^{-9}\;RIU$ was calculated for the temperature-controlled optimised measurement system. The position signals obtained by the system with an optimised angular structure between the two measurement windows are illustrated in Figure 9c, and have an average value of −0.0574 mm. The system’s jitter signal was ±0.0011 mm, and in conjunction with the sensitivity of the measurement system, a refractive index deviation of $\pm 9.93\times 10^{-10}\;RIU$ was calculated for the temperature-controlled optimised measurement system.

By comparing and analysing the position signals obtained, under the same conditions, from the seawater before and after the optimisation of the measurement system, it can be seen that the jitter signal was reduced from ±0.0022 mm to ±0.0011 mm, while the refractive index’s deviation had a reduction of $1.09\times 10^{-8}\;RIU$. The experimental results thus show that the seawater refractive index was effectively reduced by adjusting the measurement system’s optical path and structure. A comparison of the seawater refractive index measurement errors before and after the system optimisation is shown in Table 2.

## 7. Conclusions

In this paper, we theoretically analysed the causes of long jitter signals during seawater refractive index measurements and quantified the influencing factors. Through this analysis, we noticed that the angle between the two windows in the measurement area caused great errors in the measurement results, which could effectively be reduced by controlling this angle to within 2.06°. At the same time, the factors affecting the position-sensitive detector’s measurement accuracy were analysed, with the background light, the photosensitive surface size, and the temperature changes in the working environment causing its reduction. To address these factors, we added a narrow-band filter with a bandwidth of 0.9 nm in front of the photosensitive surface during the construction of the measurement system to filter out any stray light from other than the signal light source. Then, to ensure the measuring range of the seawater refractive index, a position-sensitive detector with a photosensitive surface size of 4 mm×4 mm was selected. Finally, to reduce the temperature variation in the working environment, the measurement system was partitioned. To validate the optimised test system’s error range, standard seawater samples under the same conditions were measured. The system’s jitter signal was reduced from 0.0022 mm to 0.0011 mm, while the refractive index’s deviation was reduced by $1.09\times 10^{-8}\;RIU$. The experimental results show that the seawater refractive index was effectively reduced by adjusting the measurement system’s optical path and structure. The proposed PSD-based seawater refractive index measurement system has the potential for in situ detection. In future work, this system will be applied to measure the refractive index of different marine environments. Since this system’s refractive index measurement sensitivity was higher than the refractive index change caused by the wake of underwater vehicles, it could also be used in various applications related to underwater vehicle wake measurement, as well as seawater refractive index measurements, such as the motion state monitoring of underwater navigation targets, including AUVs and ROVs.

## Figures and Tables

**Figure 1 sensors-24-04564-f001:**
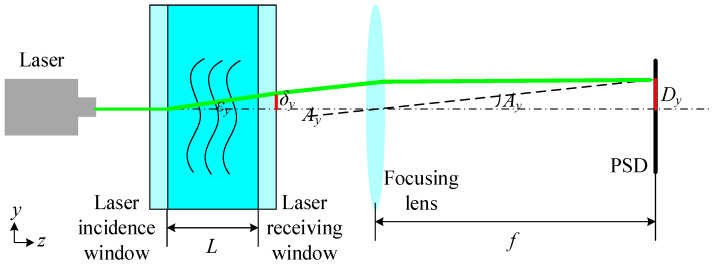
Schematic diagram of a position-sensitive detector-based seawater refractive index measurement system.

**Figure 2 sensors-24-04564-f002:**
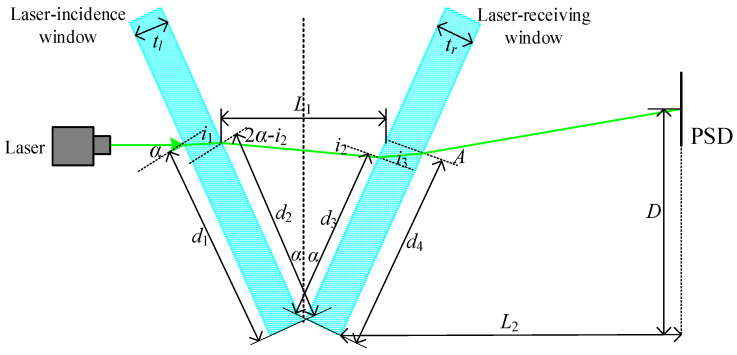
Optical path transmission process.

**Figure 3 sensors-24-04564-f003:**
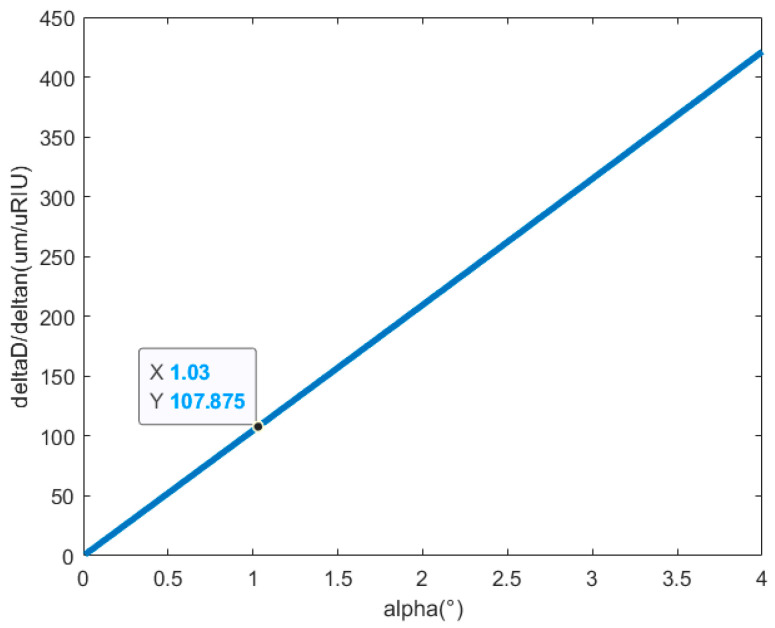
The relationship between the refractive index measurement system’s error and the angle between the two glass windows in the measurement area.

**Figure 4 sensors-24-04564-f004:**
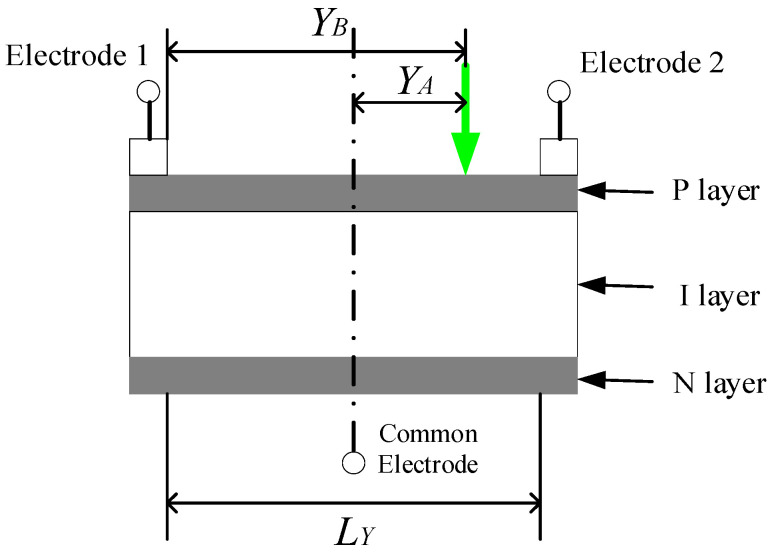
The operating principle of a position-sensitive detector.

**Figure 5 sensors-24-04564-f005:**
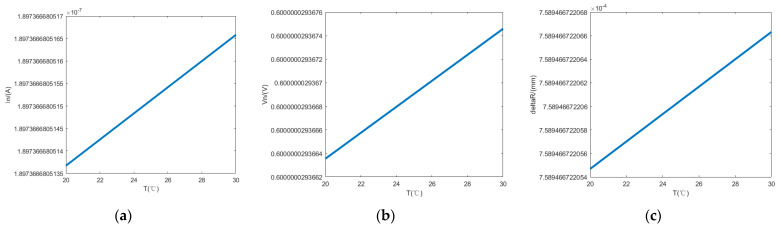
The relationship between the operating temperature and the PSD’s (**a**) noise current and (**b**) noise voltage. (**c**) The effect of the operating temperature on the PSD’s positional resolution.

**Figure 6 sensors-24-04564-f006:**
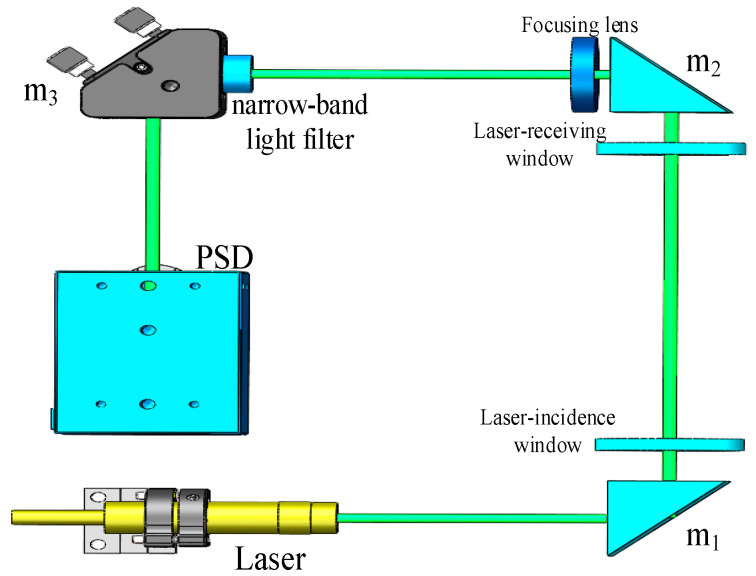
Optical path diagram of a seawater refractive index measurement system after optimisation.

**Figure 7 sensors-24-04564-f007:**
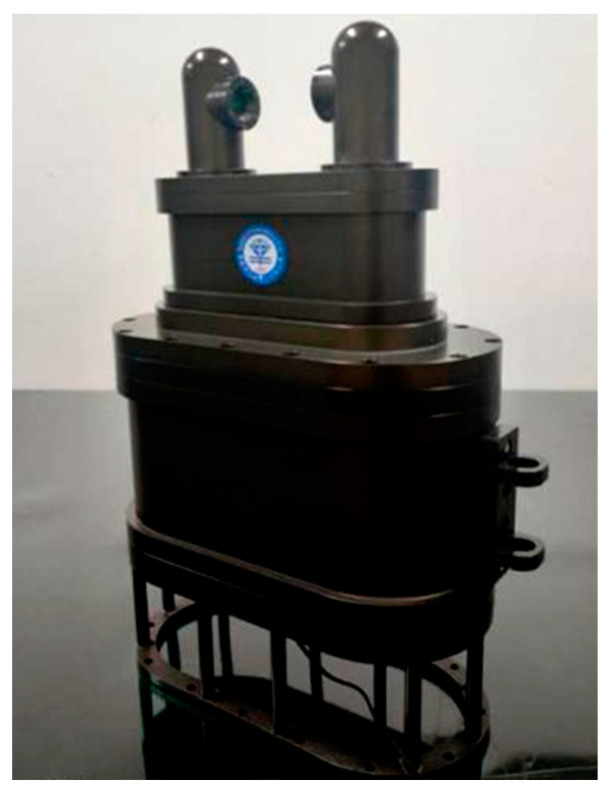
Optimised seawater refractive index measurement system.

**Figure 8 sensors-24-04564-f008:**
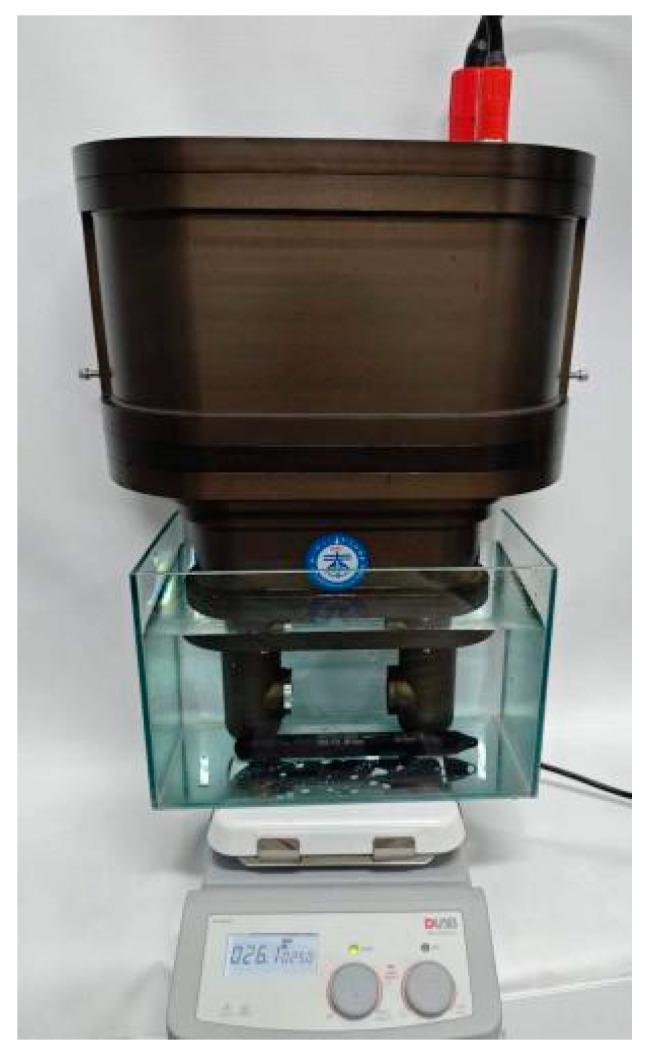
Experimental stability setup for the seawater refractive index measurement system.

**Figure 9 sensors-24-04564-f009:**
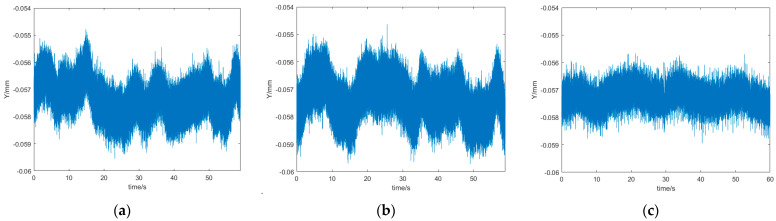
Position signals measured by the seawater refractive index measuring system (**a**) before and (**b**) after controlling the PSD’s working temperature by structural partitioning and (**c**) with an optimised angular structure between the two measurement windows.

**Table 1 sensors-24-04564-t001:** The parameters of the standard salinity seawater employed in the context of the stability experiments.

Salinity (PSU)	Pressure (dbar)	Temperature (°C)	Refractive Index (RIU)
35.003	10.1325	26.00	1.3381096

**Table 2 sensors-24-04564-t002:** A comparison of the seawater refractive index measurement errors before and after the system optimisation.

Status of the System	Laser Spot Position (mm)	Jitter Signal (mm)	Refractive Index Deviation (RIU)
Pre-optimisation	−0.0571	±0.0022	$\pm 1.19\times 10^{-8}$
After controlling the temperature	−0.0575	±0.0018	$\pm 7.944\times 10^{-9}$
After optimisation	−0.0574	±0.0011	$\pm 9.93\times 10^{-10}$

## Data Availability

The data are contained within the article.

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
