# Peer review of "Measurement Error Analysis of Seawater Refractive Index: A Measurement Sensor Based on a Position-Sensitive Detector"

_sensors, 2024, doi:10.3390/s24144564_

Round 1

Reviewer 1 Report

Comments and Suggestions for Authors

The article presents an analysis of the sources of errors in measuring the refractive index of water using PSD. The measurement of the refractive index occurs due to the bending of a laser beam when it passes through a transparent container containing a sample of seawater. According to the authors, after a rich theoretical analysis, the angular deviation due to the refraction of the laser beam is one of the major sources of error in measuring the refractive index of seawater. Furthermore, the different types of noise that can affect measurements are highlighted, such as thermal noise and noise from the operational amplifier used in the data acquisition system.

The methodology initially used promotes four refractions of the laser beam to the PSD. But the authors promoted improvements and simplifications for the optical path of the light beam used in the instrumentation. The results obtained with the optimized measurement system show improvements in the measurement of the refractive index of seawater, with a significant reduction in the jitter observed in the system readings.

The work is well written, with good illustrations and a rigorous treatment of the equations throughout the text. As a contribution, it is an important approach to refractive index measurement systems based on the technique highlighted in the work, based on PSD. However, I have the following questions and suggestions:

1) Did the authors observe any cavity effects in the experimental setup initially used? The laser beam passes through an optical path with a refractive index of 1.3 between walls with a refractive index of approximately 1.5.

2) A laser that emits at a wavelength of 532 nm was used. The authors cited light scattering and absorption phenomena as justification for choosing this laser. However, could you comment on what the impact would be with the use of another laser? For example, with a wavelength in the infrared spectrum, like 1550nm.

3) Throughout the text, the authors mention a quantity of the order of 10-9 RIU as the system's sensitivity (for example, see line 405 of the article). However, the way it is approached indicates that it functions as a resolution of the refractive index measurement system. Would it be this? If not, could you make it clearer in the text?

4) Could the optimized instrumentation be applied in the field or just to characterize samples in the laboratory? This could be highlighted in the text.

Comments on the Quality of English Language

The text is well written. No recommendations on using the English language.

Reviewer 2 Report

Comments and Suggestions for Authors

A) General Feedback:

This paper presents a valuable contribution to the field of seawater refractive index measurement which is a cruical parameter in ocean observation-studies. The authors have conducted a thorough theoretical analysis of error sources in a position-sensitive detector (PSD) based measurement system and implemented practical optimizations to enhance measurement accuracy.

1. Some of the strengths of the manuscript:

- Experimental Validation: The research includes experimental studies that validate the theoretical findings. e.g., the experiments demonstrate that the jitter signal and refractive index deviation have been significantly reduced after optimizing the system, showcasing practical improvements.

- High sensitivity: The system achieves a sensitivity of 9.93 × 10-9 RIU.

- Detailed error analysis: The paper provides a thorough theoretical analysis of possible sources of error sources and tried to quantify those influencing factors.

- Practical improvements: The authors implement and test specific optimizations to reduce measurement errors. It is shown that the optimization efforts have led to a marked reduction in the jitter signal, indicating a substantial enhancement in measurement precision.

2. Now, some of the limitations of the manuscript:

- Limited real-world testing: The study appears to focus on laboratory conditions rather than extended testing in actual marine environments.

- Potential long-term stability issues: The paper doesn't address long-term stability in seawater conditions.

- Temperature dependence: While addressed, temperature effects may still require further compensation in real-world applications.

- Narrow Range for Optimal Angles (?)

B) Specific Feedback/Comments:

1. It would be beneficial to include more details on the experimental setup and procedures used to validate the theoretical predictions (description of the glass water cylinder, calibration procedure, specs of used Laser, DAQ system etc. are missing; possible to include these?) Also, how often would recalibration be necessary in a practical deployment scenario?

2. While the laboratory results are promising, how do you envision this system performing in actual marine environments? The experiments conducted under controlled laboratory conditions, might not fully replicate real-world scenarios. Have you conducted any field tests or simulations of more challenging conditions?
Environmental variables, such as salinity variations or biological fouling in seawater could also impact the measurement accuracy but are not discussed in detail​.

3. Given the corrosive nature of seawater, how would you anticipate system's performance and stability over longer periods? Are there any plans for long-term durability testing?

4. Is it possible (it might actually strengthen the paper) to include a more detailed comparison of this PSD-based system with other current methods for measuring seawater refractive index; e.g. in terms of accuracy, cost, and ease of deployment etc. etc. or any other factors, authors find useful?

5.
-All the figures and figure-captions need to be aligned centered!
-Can figures 5 and 9 be aligned in a better alternative way?
-Also, in the description while referring the figures, sometimes 'F' was capitalized and sometime it was small! Please maintain any one convention.

6. The study finds that the angle between the laser incident and receiving windows requires to be kept within a very narrow range of 2.06° to maintain measurement sensitivity. Won't this precision be challenging to carry on consistently in practical applications (can lead to potential measurement errors)?

7. As a suggestive note- Adding a detailed table summarizing the sensitivity analysis and measurement error data for quick reference will help readers quickly understand the key findings related to the measurement error and sensitivity changes​.

8. The manuscript mentions that the research did not receive external funding, which might imply limited resources. Could this possibily affect the breadth of the experimental setups or the ability to conduct long-term studies and large-scale validations in near future?

9. Is it possible for you to discuss briefly- what do you see as the next steps in developing and validating this technology for practical use in marine science and/or related other application fields?

Comments on the Quality of English Language

While the overall structure and content of the paper are good, there are some instances where the English language could be improved for clarity and precision.

Specific areas for improvement include:

- Some sentences have been constructed in a overly complex way, making them less comprehensible. Breaking these into shorter and more direct statements would improve readability. (e.g. refer abstrct and introduction section)

- The grammatical errors throughout the text need correction. These include issues with subject-verb agreement, incorrect use of articles, and improper tense usage. (e.g. section 6.1)

- Paragraph organization: In the Results section, a few paragraphs contain a mix of methodology and results. For example, the paragraph starting with "The position signal measured by the seawater refractive index measurement system before..." combines experimental setup details with results. Separating these would improve clarity.

- A thorough proofreading is needed to catch minor errors in spelling, punctuation, and formatting.
e.g.
"Segmented optimisation BP networks and multilayer conjugate gradient optimisation algorithms are applied for nonlinear correction."
This sentence's context and connection couldn't be understood with the surrounding text, any possible editing oversight?
